# Administration of Bevacizumab and the Risk of Chronic Kidney Disease Development in Taiwan Residents: A Population-Based Retrospective Cohort Study

**DOI:** 10.3390/ijms25010340

**Published:** 2023-12-26

**Authors:** Lon-Fye Lye, Ruey-Hwang Chou, Tsai-Kun Wu, Wu-Lung Chuang, Stella Chin-Shaw Tsai, Heng-Jun Lin, Fuu-Jen Tsai, Kuang-Hsi Chang

**Affiliations:** 1Department of Medical Research, Tungs’ Taichung MetroHarbor Hospital, Taichung 435, Taiwan; lonfyelye@gmail.com; 2Graduate Institute of Biomedical Sciences, China Medical University, Taichung 404, Taiwan; rhchou@mail.cmu.edu.tw; 3Center for Molecular Medicine, China Medical University Hospital, Taichung 404, Taiwan; 4Department of Medical Laboratory and Biotechnology, Asia University, Taichung 413, Taiwan; 5Division of Renal Medicine, Tungs’ Taichung MetroHarbor Hospital, Taichung 435, Taiwan; jonesannkimo@yahoo.com.tw; 6Department of Post Baccalaureate Medicine, National Chung Hsing University, Taichung 402, Taiwan; 7Department of Internal Medicine, Division of Endocrinology and Metabolism, Changhua Christian Hospital, Changhua 500, Taiwan; tmaker@gmail.com; 8Department of Internal Medicine, Division of Endocrinology and Metabolism, Lukang Christian Hospital, Changhua 505, Taiwan; 9Department of Otolaryngology, Tungs’ Taichung MetroHarbor Hospital, Taichung 435, Taiwan; tsaistella111@gmail.com; 10Rong Hsing Research Center for Translational Medicine, College of Life Sciences, National Chung Hsing University, Taichung 402, Taiwan; 11Management Office for Health Data, China Medical University Hospital, Taichung 404, Taiwan; T38958@mail.cmuh.org.tw; 12College of Medicine, China Medical University, Taichung 404, Taiwan; 13School of Chinese Medicine, College of Chinese Medicine, China Medical University, Taichung 404, Taiwan; d0704@mail.cmuh.org.tw; 14Department of Medical Research, China Medical University Hospital, China Medical University, Taichung 404, Taiwan; 15Division of Medical Genetics, China Medical University Children’s Hospital, Taichung 404, Taiwan; 16Department of Biotechnology and Bioinformatics, Asia University, Taichung 413, Taiwan; 17Center for General Education, China Medical University, Taichung 404, Taiwan; 18General Education Center, Jen-Teh Junior College of Medicine, Nursing and Management, Miaoli 356, Taiwan

**Keywords:** bevacizumab, chronic kidney disease (CKD), angiogenesis, vascular endothelial growth factor (VEGF)

## Abstract

Vascular endothelial growth factor (VEGF) plays a significant role as a pro-angiogenic and pro-permeability factor within the kidney. Bevacizumab is a pharmaceutical monoclonal anti-VEGF antibody that inhibits the growth of new blood vessels, which blocks blood supply and thereby restricts tumor growth. Thus, we conducted a nationwide study to explore the risk of chronic kidney disease (CKD) development in Taiwan residents after bevacizumab therapy. We drew data from the extensive National Health Insurance Research Database (NHIRD), which encompasses data from >99% of Taiwan’s population from 1995 onwards. Individuals who received bevacizumab between 2012–2018 were identified as the bevacizumab cohort, with the index date set at the first usage. We randomly selected dates within the study period for the control group to serve as index dates. We excluded patients with a history of CKD prior to the index date or those <20 years old. In both cohorts, patients’ propensity scores matched in a 1:1 ratio based on sex, age, index year, income, urbanization level, comorbidities, and medications. We found patients treated with bevacizumab had a significantly higher risk of contracting CKD than patients without bevacizumab (adjusted hazard ratio = 1.35, 95% confidence interval = 1.35–1.73). The risk of CKD was 1.35-fold higher in participants with bevacizumab treatment than those in the control group. These findings suggest that close monitoring of CKD development after bevacizumab administration is needed.

## 1. Introduction

Chronic kidney disease (CKD) is characterized by kidney damage or a glomerular filtration rate (GFR) persistently below 60 mL/min/1.73 m^2^ for three months or longer [1,2]. Pre-end stage renal disease (pre-ESRD) causes dangerous levels of fluid, electrolytes, and waste to build up in the patient’s body [3]. Endothelial dysfunction is a significant factor contributing to the increased cardiovascular risk associated with CKD [4]. Angiogenesis, the generation of new blood vessels, is an essential physiological process that is deregulated in various pathological conditions, including cancer [5,6]. Anti-angiogenesis therapies, including vascular endothelial growth factor (VEGF) targeting agents, offer an alternative strategy to conventional chemotherapy; they inhibit the growth of new blood vessels, which blocks blood supply and thereby restricts tumor growth [7]. Bevacizumab is a humanized monoclonal antibody directed against VEGF [8,9,10,11]. The drug selectively binds to circulating VEGF and inhibits the binding of VEGF to its cell surface receptor, thereby repressing angiogenic activity [12,13,14]. It is used for various medical conditions and is associated with recognized renal side effects, particularly proteinuria and acute kidney injury (AKI) [15]. A previous study showed that out of the 1506 patients who were administered bevacizumab, the renal events associated with this medication were as follows: 125 instances of acute kidney injury (AKI), which accounted for 8.3% of cases; 110 occurrences of new-onset proteinuria, representing 7.3% of patients; and new-onset CKD in 112 individuals, amounting to 7.4% of the total [16]. However, the authors were unable to calculate the relative risk of CKD because no control group was available for comparison. Hence, limited information is available regarding the extended, or long-term, consequences of its renal adverse effects.

We hypothesized that the anti-angiogenesis effect of bevacizumab may impede the generation of new vessels after renal injury, leading to CKD. Thus, we conducted a nationwide study to explore the risk of CKD development in Taiwan residents after bevacizumab therapy.

## 2. Results

The demographic characteristics of the study group are listed in Table 1. The dataset included 13,888 individuals within the control group, classified as the non-bevacizumab category, and 13,888 patients within the case group, who received bevacizumab. After propensity score matching, both groups exhibited similar baseline distributions for specific characteristics, such as sex, income, and the comorbidity of alcoholism. The mean follow-up time was 1.7 ± 1.3 years and 2.3 ± 2.0 years with and without bevacizumab, respectively.

In the bevacizumab cohort, 58% were men and the mean age was 62 ± 13 years; 48% were classified as medium income and 53% were categorized as high urbanization level. The three most prevalent comorbidities were hypertension (HTN) (47%), dyslipidemia (33%), and diabetes mellitus (DM) (26%). The most frequently prescribed medications included calcium-channel blockers (CCB) (51%), diuretics (48%), alpha-adrenergic blockers (AAB) (32%), and angiotensin receptor blocker (ARB) (31%). In the non-bevacizumab group, 58% were men and the mean age was 62 ± 13 years; 50% were classified as medium income level and 52% were categorized as high urbanization level. The three most prevalent comorbidities were hypertension (48%), dyslipidemia (32%), and DM (26%). The most frequently prescribed medications included CCB (52%), diuretics (50%), alpha-adrenergic blockers (33%), and ARB (33%).

Table 2 presents the risk of CKD associated with bevacizumab. Patients treated with bevacizumab had a significantly higher risk of developing CKD than patients without bevacizumab (adjusted HR [aHR] = 1.35, 95% CI = 1.18–1.55). Additionally, other risk factors for CKD included male sex (aHR = 1.18, 95% CI = 1.01–1.38), age between 50 and 64 years (aHR = 1.42, 95% CI = 1.06–1.89), age >65 years (aHR = 2.06, 95% CI = 1.54–2.76), DM (aHR = 1.84, 95% CI = 1.58–2.14), angiotensin converting enzyme inhibitors (ACEI) (aHR = 1.18, 95% CI = 1.00–1.38), alpha-adrenergic blockers (aHR = 1.30, 95% CI = 1.11–1.53), ARB (aHR = 1.23, 95% CI = 1.03–1.47), CCB (aHR = 1.25, 95% CI = 1.02–1.52), and diuretics (aHR = 1.76, 95% CI = 1.51–2.04), all of which also displayed a significantly increased risk of CKD.

Furthermore, we conducted a stratified analysis, as presented in Table 3. Compared to control patients, both males and females of bevacizumab faced a greater risk of CKD. Across all age and income groups, the risks of CKD associated with bevacizumab treatment were higher than control groups. When stratified by comorbidities and medications, patients within the bevacizumab cohort with DM, dyslipidemia, HTN, COPD, alpha-adrenergic blocker usage, ARB usage, beta blocker usage, CCB usage, diuretic usage, and other anti-HTN drugs usage exhibited an increased risk of CKD compared to the non-bevacizumab cohort in these same categories.

Figure 1 shows that the cumulative incidence curves for CKD in the bevacizumab group were significantly higher than those in the comparison group (log-rank test *p* < 0.001).

## 3. Discussion

The molecular mechanisms involved in the pathophysiology and progression of CKD and associated complications have been documented in many studies [17]. CKD encompasses a range of diverse disorders that impact both the structure and function of the kidneys [18]. CKD progresses with the accumulation of extracellular matrix, leading to end-stage renal disease. In general, diabetes and high blood pressure are the more common causes of CKD. Other risk factors affecting CKD include heart disease, obesity, a family history of CKD, and inherited kidney disorders [19]. VEGF is the essential growth factor in angiogenesis and is a well-characterized contributor to angiogenesis. VEGF-A and VEGF-C secreted by podocytes are essential for maintaining the physiological functions of glomerular endothelial cells and podocytes via binding to their receptors, VEGFR-1 (Flt-1), VEGFR-2 (KDR/Flk-1), VEGFR-3 (Flt-4), and coreceptors, NRP (Neuropilin) 1 and NRP2 [7]. The dysregulated high circulating level of VEGF is associated with diabetic retinopathy and diabetic nephropathy [20]. VEGF mRNA level and protein expression in kidney tissue are significantly higher in diabetic patients with nephropathy than those without nephropathy. VEGF expression is closely linked to renal function and the disease progression of diabetic nephropathy and is an independent risk factor of prognostic recurrence [21].

Bevacizumab, a VEGF inhibitor, is a well-known anti-angiogenesis medication [8,9,10,11]. Furthermore, disrupting VEGF signaling activates CD4^+^ and CD8^+^ T cells, enhances the antigen presentation function of dendritic cells, facilitates the cytotoxicity of macrophage, and activates the complement cascade [7]. Numerous prior investigations have sought to elucidate the potential pathophysiological process behind renal damage and proteinuria associated with bevacizumab [22,23,24]. This may be attributed to a high dose of bevacizumab [15,25]. The intravitreal injection of bevacizumab in the patients with diabetes increases the concentrations of microalbumin in the urine and the ratio of urinary protein to creatinine [26], and is associated with renal function decline among patients with diabetic retinopathy and advanced CKD [27]. Renal damage occurs in the diabetic macular edema patients with an increase in microalbuminuria and a decrease in the glomerular filtration rate (eGFR) after intravitreal anti-VEGF therapy, regardless of the number of injections or the type of VEGF inhibitors [28]. Patients with extremely high or low eGFR have greater persistent eGFR decline during the long-term intravitreal administration of anti-VEGF treatment for two years in the patients with diabetic macular edema [29]. Gan et al. reported that a patient with macular edema secondary to central retinal vein occlusion has membranoproliferative glomerulonephritis after intravitreal vascular growth factor inhibitor injections [30]. However, there is no clinically significant change in the serum markers of renal function during the short-term intravitreal treatment of anti-VEGF drugs for 1 month in the patients with neovascular age-related macular degeneration [31]. In addition, there is no association between intravitreal anti-VEGF treatment and renal adverse events based on the data from the FDA’s Adverse Event Reporting System (FARES) database [32].

This study focused on CKD development after the administration of bevacizumab. While VEGF targeting offers substantial benefits in the treatment of various tumors, it can also lead to a range of adverse effects, such as wound non-healing, HTN, embolism, bleeding, and phlebitis [33,34,35,36], which typically results in treatment discontinuation [37]. Uys et al. reported that continuous bevacizumab use led to renal thrombotic microangiopathy (TMA) [38]. In addition, several cases of renal dysfunction and proteinuria have been reported after exposure to bevacizumab [26,27,28,30,32]. A previous study suggested that TMA was associated with a risk of CKD [39]. Based on this nationwide study, we conclude that there is a risk of CKD development after the administration of bevacizumab regardless of sex, most comorbidities, income, and medications. A possible mechanism linking bevacizumab with CKD is vascular microangiopathy caused by bevacizumab [40,41,42]. A study of kidney injury mechanisms in rats indicated that levels of microalbumin, cystatin C, serum creatinine, and blood urea nitrogen in the bevacizumab group were markedly elevated compared to those in the normal control group [15]. Moreover, there is a potential for VEGF inhibitors to exert a toxic effect on podocytes, contributing to their damaging effects [43]. One of the mechanisms of endothelial dysfunction involves less promotion of nitric oxide, leading to higher vascular tone and vasoconstriction, potentially promoting microvascular thrombi formation. Furthermore, the interaction between VEGF-A and VEGFR2 mainly expressed by glomerular podocytes and on the surface of endothelial cells, respectively, is important for the maintenance of the normal function of glomerular and tubular epithelial cells in the kidney [44]. The disruption of VEGF-A/VEGFR2 signaling by VEGF inhibitors, such as bevacizumab, increases RelA (NF-kB transcription factor, also known as p65) and decreases c-MIP (C-Maf-inducing protein) to promote the development of MTA, leading to the activation of the pro-inflammatory cascade and endothelial injuries [45]. The aforementioned nephrotoxicity and injuries in the kidney might contribute to bevacizumab associated risk of CKD.

Although this nationwide study had a well-structured design, there remain some limitations. First, we enrolled control participants by propensity score matching based on sex, age, index year, income, urbanization level, comorbidities, and medications. Thus, the prevalence of comorbidities in the control group may not be representative of the general population. This may have resulted in an underestimation of the risk of CKD development and the non-significant association found between CKD and most comorbidities (Table 2).

Second, different medical conveniences in urbanization may affect willingness to seek medical treatment and reduce the prevalence of comorbidities. This study did not enroll participants from remote areas. Nevertheless, the National Health Insurance (NHI) program covers >99% of Taiwan residents and reduces the gap in medical convenience in remote areas by providing free medical care [46,47], thus minimizing surveillance bias. Third, in the initial phases of CKD (stages 1–2), there are no clinical symptoms, and patients experience a lack of noticeable signs [19]. In addition, its characteristic is a prolonged latent phase [48,49]. Therefore, CKD risk may be underestimated, as we were unable to enroll patients who were diagnosed with CKD outside of the study period. Fourth, as bevacizumab is primarily used as a cancer treatment, establishing whether CKD results from the drug itself or the underlying cancer remains a complex task. Further research is necessary to delineate the connection between medications and CKD.

In conclusion, closely monitoring for CKD development after the administration of bevacizumab is urgently required. Accelerating basic research in this area in order to enable early diagnosis of CKD is critical. Awareness of these complications should be further elaborated to assess the safety profile of bevacizumab administration.

## 4. Materials and Methods

### 4.1. Data Source

Our analysis draws from the extensive National Health Insurance Research Database (NHIRD), which encompasses data from >99% of Taiwan’s population spanning from 1995 onwards. The database encompasses detailed medical records from outpatient visits and hospital admissions, inclusive of diagnostic codes and prescription medications. To safeguard individual privacy, the NHIRD employs rigorous anonymization protocols. While the database contains comprehensive medical information, individual identities are meticulously protected. Researchers, healthcare professionals, and analysts can access de-identified data, ensuring confidentiality and compliance with ethical standards. The International Classification of Diseases, Ninth Revision, Clinical Modification (ICD-9-CM), and Tenth Revision, Clinical Modification (ICD-10-CM), were used for disease classification. This study has been approved by the Research Ethics Committee at the China Medical University Hospital (CMUH110-REC3-133-CR-2).

### 4.2. Study Subjects

The population we included in this study were cancer patients (ICD-9-CM: 140–208; ICD-10-CM: C00-C99). Individuals who received bevacizumab between 2012–2018 were identified as the bevacizumab cohort. Within this cohort, we designated the index date as the date of their initial bevacizumab usage. For the control group, we randomly selected dates within the study period to serve as index dates. We excluded patients with a history of CKD prior to the index date or those who were <20 years old. The two groups were paired using a 1:1 propensity score matching approach, considering factors such as gender, age, index year, urbanization, DM, dyslipidemia, HTN, stroke, CAD, alcoholism, COPD, ACEI, AAB, ARB, beta-blocker, CCB, diuretics, and other anti-HTN drugs.

### 4.3. Main Outcome and Relevant Variables

The primary focus of this research was to determine the incidence of CKD (ICD-9-CM: 580; ICD-10-CM: N18.4, N18.5, N18.6, and N18.9) for at least two outpatient claims or one inpatient claim. Throughout the study, all enrolled patients were followed up from the index date until the earliest occurrence of one of three events: (1) the onset of CKD, (2) death, or (3) the conclusion of the observation period on 31 December 2019. In our study, we chose the following comorbidities: diabetes mellitus (DM; ICD-9-CM: 250; ICD-10-CM: E08-E13), dyslipidemia (ICD-9-CM: 272; ICD-10-CM: E71.30, E75.21, E75.22, E75.24, E75.3, E75.5, E75.6, E77, E78.0, E78.1, E78.2, E78.3, E78.4, E78.5, E78.6, E78.70, E78.79, E78.8, and E78.9), hypertension (HTN; ICD-9-CM: 401–405; ICD-10-CM: I10-I13, I15, and N26.2), stroke (ICD-9-CM: 430–438; ICD-10-CM: I60-I69), coronary artery disease (CAD; ICD-9-CM: 410–414; ICD-10-CM: 20-I25), alcoholism (ICD-9-CM: 291, 303, 305.0, 571.0–571.3, 790.3, V11.3, and V79.1; ICD-10-CM: F10, K70, R78.0, and Z65.8), and chronic obstructive pulmonary disease (COPD; ICD-9-CM: 490–496, and 504–506; ICD-10-CM: J40-J47, and J64-J68). Furthermore, we conducted an analysis of common medication usage, including angiotensin-converting enzyme inhibitors (ACEI), alpha-adrenergic blockers, angiotensin II receptor blockers (ARB), beta blockers, calcium channel blockers (CCB), diuretics, and other anti-HTN drugs.

### 4.4. Statistical Analysis

We calculated the frequency in percentages for each categorical variable. Continuous variables were presented as the mean with standard deviation (SD). To assess descriptive statistics between the case and control groups, we used standardized mean difference (SMD) analysis. A negligible imbalance in potential confounders between the two groups was indicated by SMD < 0.1. In the univariate Cox proportional hazards model, we computed the hazard ratios (HR) along with their corresponding 95% confidence intervals (95% CI). The multivariate Cox proportional hazards model accounted for factors such as age, sex, comorbidities, and medications. We defined a *p*-value < 0.05 in two-tailed tests as being statistically significant. All analyses were performed using SAS 9.4 (SAS Institute, Cary, NC, USA).

## 5. Conclusions

Accelerating basic research in this area in order to enable the early diagnosis of CKD is critical. Awareness of these complications should be further elaborated to assess the safety profile of bevacizumab administration.

## Figures and Tables

**Figure 1 ijms-25-00340-f001:**
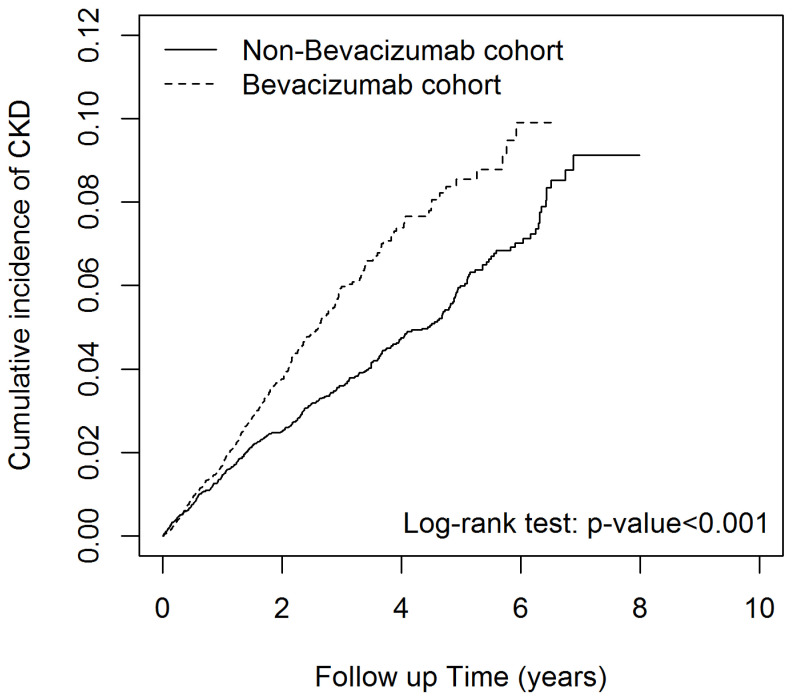
Cumulative incidence of CKD compared between patients with and without bevacizumab.

**Table 1 ijms-25-00340-t001:** Comparison of demographic characteristics and comorbidities between patients treated with bevacizumab and controls.

Covariates		Non- Bevacizumab (*n* = 13,888)	Bevacizumab (*n* = 13,888)	SMD
Sex				<0.001
	Female	42%	42%	
	Male	58%	58%	
Age				
	20–49	16%	17%	0.009
	50–64	43%	42%	0.013
	65+	41%	41%	0.006
	mean ± SD	62 ± 13	62 ± 13	0.002
Insurance fee				
	<20,000	31%	33%	0.048
	20,000–39,999	50%	47%	0.050
	40,000+	19%	19%	0.006
Urban				
	Low	8%	8%	<0.001
	Medium	39%	39%	0.007
	High	52%	53%	0.006
Comorbidities				
	DM	26%	26%	0.011
	Dyslipidemia	32%	33%	0.016
	HTN	48%	47%	0.019
	Stroke	11%	11%	0.012
	CAD	15%	15%	0.001
	Alcoholism	2%	2%	0.016
	COPD	22%	21%	0.028
Medication				
	ACEI	19%	19%	0.004
	AAB	33%	32%	0.019
	ARB	33%	31%	0.024
	Beta-blocker	20%	20%	0.009
	CCB	52%	51%	0.025
	Diuretics	50%	48%	0.042
	Others	6%	6%	0.021
Follow up years	mean ± SD	2.2 ± 2.0	1.7 ± 1.3	0.342

SMD, standardized mean difference; income: low is <USD 700/month, medium is USD 701–1400/month, high is USD 1400+/month; DM, diabetes mellitus; HTN, hypertension; CAD, coronary artery disease; COPD, chronic obstructive pulmonary disease; ACEI, angiotensin converting enzyme inhibitors; AAB, alpha-adrenergic blockers; ARB, angiotensin receptor blocker; CCB, calcium-channel blockers; others, other anti-HTN drugs.

**Table 2 ijms-25-00340-t002:** Hazard ratios and 95% confidence intervals of CKD development.

Covariates		CKD		aHR	95% CI
		*n*	IR		
Bevacizumab					
	No	413	12.8	1.00	(reference)
	Yes	456	18.9	1.35	(1.18, 1.55)
Sex					
	Female	308	12.1	1.00	(reference)
	Male	561	18.1	1.18	(1.01, 1.38)
Age					
	20–49	58	5.6	1.00	(reference)
	50–64	300	11.7	1.42	(1.06, 1.89)
	65+	511	25.0	2.06	(1.54, 2.76)
Income					
	Low	281	16.5	1.00	(reference)
	Medium	432	15.3	0.92	(0.79, 1.08)
	High	156	13.9	0.90	(0.74, 1.10)
Urban					
	Low	68	15.6	1.00	(reference)
	Medium	358	16.2	1.08	(0.83, 1.40)
	High	443	14.8	1.05	(0.81, 1.36)
Comorbidities					
	DM	403	30.2	1.84	(1.58, 2.14)
	Dyslipidemia	388	22.1	0.92	(0.79, 1.07)
	HTN	613	24.5	1.20	(0.96, 1.50)
	Stroke	136	28.5	1.12	(0.92, 1.35)
	CAD	190	25.0	0.87	(0.73, 1.04)
	Alcoholism	17	20.1	1.04	(0.64, 1.69)
	COPD	222	20.0	0.97	(0.83, 1.14)
Medication					
	ACEI	283	29.7	1.18	(1.00, 1.38)
	AAB	408	25.1	1.30	(1.11, 1.53)
	ARB	447	28.2	1.23	(1.03, 1.47)
	Beta-blocker	205	19.9	1.08	(0.92, 1.28)
	CCB	617	23.4	1.25	(1.02, 1.52)
	Diuretics	570	24.3	1.76	(1.51, 2.04)
	Others	88	31.3	1.20	(0.95, 1.50)

*n*, number of patients with CKD; IR, incidence rate ratio (1000 person-years); aHR, adjusted hazard ratio; CI, confidence interval; income: low is <USD 700/month, medium is USD 701–1400/month, high is USD 1400+/month; DM, diabetes mellitus; HTN, hypertension; CAD, coronary artery disease; COPD, chronic obstructive pulmonary disease; ACEI, angiotensin converting enzyme inhibitors; AAB, alpha-adrenergic blockers; ARB, angio-tensin receptor blocker; CCB, calcium-channel blockers; others, other anti-HTN drugs.

**Table 3 ijms-25-00340-t003:** Hazard ratios and 95% confidence intervals of CKD development with and without bevacizumab stratified by age, comorbidities, and medication.

Covariates		Non- Bevacizumab		Bevacizumab		aHR	95% CI
		*n*	IR	*n*	IR		
Sex							
	Female	148	9.74	160	15.47	1.41	(1.12, 1.77)
	Male	265	15.48	296	21.37	1.33	(1.12, 1.57)
Age							
	20–49	18	2.89	40	9.77	2.74	(1.55, 4.85)
	50–64	141	9.54	159	14.53	1.43	(1.13, 1.80)
	65+	254	22.50	257	28.06	1.20	(1.01, 1.43)
Income							
	Low	129	13.90	152	19.70	1.29	(1.01, 1.64)
	Medium	216	13.15	216	18.28	1.27	(1.05, 1.54)
	High	68	10.30	88	18.87	1.69	(1.21, 2.35)
Urban							
	Low	35	14.28	33	17.16	1.13	(0.69, 1.85)
	Medium	174	13.66	184	19.56	1.31	(1.06, 1.63)
	High	204	11.92	239	18.58	1.40	(1.16, 1.70)
Comorbidities							
	DM	203	27.95	200	32.83	1.19	(0.97, 1.46)
	Dyslipidemia	188	19.47	200	25.36	1.28	(1.05, 1.57)
	HTN	298	21.47	315	28.27	1.30	(1.11, 1.53)
	Stroke	80	31.56	56	24.94	0.87	(0.61, 1.24)
	CAD	101	24.74	89	25.29	1.10	(0.82, 1.48)
	Alcoholism	9	22.45	8	18.00	0.98	(0.34, 2.81)
	COPD	106	17.15	116	23.62	1.38	(1.05, 1.81)
Medication							
	ACEI	148	28.71	135	30.80	1.09	(0.86, 1.39)
	AAB	197	22.23	211	28.62	1.29	(1.06, 1.57)
	ARB	210	23.87	237	33.59	1.37	(1.13, 1.66)
	Beta-blocker	98	17.16	107	23.34	1.32	(1.00, 1.76)
	CCB	292	20.14	325	27.36	1.34	(1.14, 1.58)
	Diuretics	283	22.68	287	26.23	1.16	(0.98, 1.37)
	Others	38	27.78	50	34.66	1.28	(0.83, 1.97)

*n*, number of patients with CKD; IR, incidence rate ratio (1000 person years); aHR, adjusted hazard ratio; CI, confidence interval; income: low is <USD 700/month, medium is USD 701–1400/month, high is USD 1400+/month; DM, diabetes mellitus; HTN, hypertension; CAD, coronary artery disease; COPD, chronic obstructive pulmonary disease; ACEI, angiotensin converting enzyme inhibitors; AAB, alpha-adrenergic blockers; ARB, angio-tensin receptor blocker; CCB, calcium-channel blockers; others, other anti-HTN drugs.

## Data Availability

Information is accessible through the National Health Insurance Research Database (NHIRD) provided by the Taiwan National Health Insurance (NHI) Administration. However, due to legal constraints stipulated by the Taiwan government under the “Personal Information Protection Act”, the data cannot be released to the public. If you wish to obtain the data, you may submit formal requests to the NHIRD via their website (https://dep.mohw.gov.tw/DOS/lp-2506-113.html, accessed on 14 December 2023).

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
