# Peer review of "Administration of Bevacizumab and the Risk of Chronic Kidney Disease Development in Taiwan Residents: A Population-Based Retrospective Cohort Study"

_ijms, 2023, doi:10.3390/ijms25010340_

Round 1

Reviewer 1 Report

Comments and Suggestions for Authors

This study used the extensive NHIRD, which covers a large percentage of Taiwan's population and provides detailed medical records. This allows for a comprehensive analysis of the association between bevacizumab use and CKD. Nevertheless, some issues should be addressed:

1) While the study mentions that factors such as age, sex, comorbidities, and medications were accounted for, more information on the adjustment methods would enhance the validity of the study's findings. For example, all types of medication were used more frequent in bevacizumab group (except ACEI), and were linked to a higher incidence of CKD. The real impact of all these medications could be unclear, and could impact the effect of bevacizumab on CKD occurrence.

2) The authors should define chronic kidney disease in the methods section (would increase the validity of the reported results)

3) The process of selecting controls should be more detailed (there were selected healthy control in propensity matching?)

4) As the authors stated in the limitations, it is hard to establish whether cancer was the main cause of CKD instead of bevacizumab (as it used in cancer patients). It would be useful to compare cancer patients treated with bevacizumab with those not treated with bevacizumab. It seems that cancer was not included as a confounding factor in the analysis process. It would be useful to state how many patients suffered from cancer in both analyzed cohorts.

Author Response

This study used the extensive NHIRD, which covers a large percentage of Taiwan's population and provides detailed medical records. This allows for a comprehensive analysis of the association between bevacizumab use and CKD. Nevertheless, some issues should be addressed: 1) While the study mentions that factors such as age, sex, comorbidities, and medications were accounted for, more information on the adjustment methods would enhance the validity of the study's findings. For example, all types of medication were used more frequent in bevacizumab group (except ACEI), and were linked to a higher incidence of CKD. The real impact of all these medications could be unclear, and could impact the effect of bevacizumab on CKD occurrence. Reply We conducted a reanalysis as per the recommendation. In the updated findings, the disparity between the two groups diminished. 2) The authors should define chronic kidney disease in the methods section (would increase the validity of the reported results) Reply We added the definition of CKD: Chronic Kidney Disease (CKD) is characterized by kidney damage or a glomerular filtration rate (GFR) persistently below 60 mL/min/1.73 m2 for three months or longer. (Line 59-61) 3) The process of selecting controls should be more detailed (there were selected healthy control in propensity matching?) Reply We rewrote the paragraph in the section on Study subjects. The two groups were paired using a 1:1 propensity score matching approach, considering factors such as gender, age, index year, urbanization, DM, dyslipidemia, HTN, stroke, CAD, alcoholism, COPD, ACEI, AAB, ARB, beta-blocker, CCB, diuretics, and other an-ti-HTN drugs. (line 255-258) 4) As the authors stated in the limitations, it is hard to establish whether cancer was the main cause of CKD instead of bevacizumab (as it used in cancer patients). It would be useful to compare cancer patients treated with bevacizumab with those not treated with bevacizumab. It seems that cancer was not included as a confounding factor in the analysis process. It would be useful to state how many patients suffered from cancer in both analyzed cohorts. Reply We appreciate all the above suggestions and reanalysis. All of the participants were cancer patients. We added the sentence in the section on Study subjects as follows: The population we inclusion of this study were cancer (ICD-9-CM: 140-208; ICD-10-CM: C00-C99). (line 250-251)

Reviewer 2 Report

Comments and Suggestions for Authors

The subject the authors address, of CKD worsening during bevacizumab treatment is an important one and the authors’ finding of an epidemiological association is not a new finding but a useful one that should be published, confirming previous findings of others. 

I have a minor ethical concern that the authors did not refer to previous similar findings. Confirming previous findings of others is useful and important. Hopefully their omission was inadvertent or due to poor scholarship, not an attempt at hiding previous related work. English use is good but presentation is a bit sloppy and must be corrected. A few sentences are hard to understand and must be corrected.

Line 35 assumes readers know what bev is. Dont assume this. Better would be “Bevacizumab is a pharmaceutical monoclonal anti-VEGF antibody that…”

Line 40 a dash or an “and” missing “2012 2018”. So “between 2012 and 2018 were identified…”.

Line 47, would the authors think it helpful to list the actual numbers here ?

Line 54. Would it be helpful to add the common definition of CKD as GFR  below 60 ml/min/1.73M2 ?

Line 56 I don’t think we can speak of “serious CKD” in a medical or formal paper on renal function. I think you must define any discussion of CKD in terms of grade [1, 2, 3 or 4] and the implications each grade has. Remember you are addressing both non-clinical researchers as well as front line physicians. This is not a newpaper article.

Your sentence “Endothelial dysfunction is a significant factor contributing to the increased cardiovascular risk associated with CKD(2).” is a bit unclear or confusing. The relationship between CKD, endothelial dysfunction and cardiovascular disease [CVD] are indeed related but you must explain this more clearly. It isn't simple. Vascular pathology can lead to both CKD and to CVD. CKD can generate systemic vascular pathology and CVD. A fourth factor [chemotherapies, substance abuse, sepsis, etc] can generate all three, CKD, CVD and systemic vasculopathy. An example of this fourth potential relationship was discussed in H. Chen et al below. You must find a way to simply but clearly say this.

Sometimes there is a space between the last word and the references and sometimes there isn’t [VEGF(7-10)]. Fix this such that there is always a space.

Lines 65 to 69 should be a separate paragraph. 

Line 76, this might be an idiosyncrasy of mine, but giving these statistics to the hundredth place seems silly, obfuscating and scientifically wrong or misleading. The authors do not doubt those numbers in the hundredth place would be different if they used a different time frame for both cohorts or a different number in each cohort. Remember the word “significance has two meanings - 1) in statistics, and 2) meaningfulness to the reader. If I hand you a penny you are statistically richer but not meaningfully richer. If I hand a starving man two grains of rice by technical measurement he has more food than he did prior to that, a statistically meaningful difference, but not a meaningful difference to him. Likewise reporting “1.73±1.32 years“ is distracting and deceptive. Would not “1.7 years ± 1.3” be more meaningful and easier to read and absorb ? My similar complaint applies throughout this paper.

Lines 77 to 79 are example of above. The authors state “In the bevacizumab cohort, 58.3% were men, and the mean age was 61.53±12.49 years; 47.4% were classified as medium income and 52.7% were categorized as high urbanization level.”. My view is better would be “In the bevacizumab cohort, 58% were men, and the mean age was 62 ±12 years; 47% were classified as medium income, 53% were categorized as high urbanization level.” This would be a more effective communication and more importantly, IMO, more accurate. 

Line 79 starts a new subject, should therefore start new paragraph.

Line 80-81, CCB not previously defined. “alpha-blockers” is unacceptable [alpha-adrenergic blockers presumably], ARB not previously defined, DM not previously defined, 

Line 81, no ACE inhibitors ? in USA, Canada, and the EU lisinopril is commonly used in CKD. You didn’t have many lisinopril pts ?

Line 88, “contracting” incorrect word. “developing” is your intended meaning.

Lines 88 to 94, I can assure the authors that they obscure their message by this paragraph. First you use CI here and ± elsewhere. Are these not the same ? Also in this paragraph the use of hundredth place is utterly unwarranted and obscuring to your findings. I would also claim that use of tenth place is superfluous, distracting, and unwarranted by the nature of your cohort or your findings.  

Line 94 HTN not previously defined. Although most clinical people in English speaking countries will know what this stands for, non-clinicians may not and those in non-English speaking areas might not. This error of presentation again reflects the authors not carefully thinking of how their readers might be positioned.

Line 104 COPD not previously defined.

Line 92 ACEI not previously defined. 

Lines 96 to 108 must be rewritten. 

Your message will be lost when you try to present data this way in a maze of numbers. I suggest simply reduce this entire paragraph to….. 

“Furthermore, we conducted a stratified analysis as presented in Table 3. Compared to control patients, both males and females of bevacizumab faced a greater risk of CKD. Across all age and income groups, the risks of CKD associated with bevacizumab treatment were higher than control groups. When stratified by comorbidities and medications, patients within the bevacizumab cohort with DM, dyslipidemia, HTN, COPD, alpha-adrenergic blocker usage, ARB usage, beta blocker usage, CCB usage, diuretic usage, and other anti-HTN drugs usage exhibited an increased risk of CKD compared to the non-bevacizumab cohort in these same categories.”. 

There is no need to duplicate numbers in the tect that are listed in Table3.

I suggest leaving out the actual number of people in all Tables. Just list %.

SMD requires defining in Tables.

Table 2, better just list %. TWD needs defining. CI need not be defined. Would it help international readers to show TWD in USD or Euros ?

I don’t think Table 3 will be of much use to people. I am a physician-clinician and I think I am representative of that group. We are much more likely to wade through that maze of numbers if you eliminated actual numbers in each group and just listed %, HR and CI.

Line 125, no. CKD is not “CKD is a condition in which the kidneys are damaged causing impaired renal function.” CKD refers to a condition of GFR <60. Often, but not always, due to glomerular damage. 

Line 138 is incorrect. We see wound non-healing and HTN as the most common bevacizumab short term adverse events. 

Line 151, is incorrect. “poisonous effects” is not the same as damaging effects.

Line 155, 156 is bad grammar. Paragraph 155-158 must be rewritten. Perhaps say simply “others studying intraocular bevacizumab did not find reduced GFR”. It would be obvious why they did not. But some did and they too must be discussed

Line 167, I did not understand the authors’ meaning. [ Nevertheless, the National Health Insurance (NHI) program covers >99% of Taiwan residents and reduces the gap in medical convenience in remote areas by providing free medical care(37, 38), thus minimizing surveillance bias.] ? Please rephrase.

Line 165, new aspect. “Second, different medical…” start new paragraph.

Line 167 again shows the authors’ unclarity on what CKD incompases. Stage 1 and 2 are usually clinically silent but 3 and 4 are less so.

“closely” and “urgently” must be eliminated. GFR is always monitored during most cancer treatments and so will catch any reduced GFR. The authors could mention that understanding the details of how exactly bevacizumab damages endothelium might help us protect glomeruli during bev treatment. Also in cases of grade 3 CKD one might carefully balance r/b during cancer use. Not rare is sign/symptom improvement in cancer but no prolonging of survival with bev use is seen. One might then consider which is worse, bev or the signs/symptoms one is trying to relieve. 

References A to J must be discussed and included in Discussion Section. A diagram [a cartoon or schematic drawing] would be helpful when discussing these papers. Also the authors should review the way that these authors presented their data. Note for example that Fang et al was easy to read, omitted the hundredth place in data, and was good example of effective medical communication.

Reference 1 is intolerable in a medical scientific research article.

The entire reference list is too old, it must be redone and the text changed correspondingly to reflect current understandings of endothelium, bevacizumab and CKD. The authors’ physiology discussions are inadequate, out-of-date and must be corrected.

-------------------------------------------------------------------------

A. Li J, Li XL, Li CQ. Immunoregulation mechanism of VEGF signaling pathway inhibitors and its efficacy on the kidney. Am J Med Sci. 2023 Sep10:S0002-9629(23)01330-7. doi: 10.1016/j.amjms.2023.09.005. 

B. Zhang J, Zhu H, Tang Q, Wu J. Correlation Analysis of Vascular Endothelial Growth Factor Level with Clinicopathological Features and Prognosis in Patients with Diabetic Nephropathy: A Biopsy-based Study. Cell Mol Biol (Noisy-le-grand). 2023 Aug 31;69(8):192-197. doi: 10.14715/cmb/2023.69.8.29. 

C. Fang YC, Lai IP, Lai TT, Chen TC, Yang CH, Ho TC, Yang CM, Hsieh YT. Long-Term Change in Renal Function After Intravitreal Anti-VEGF Treatment for Diabetic Macular Edema: A 2-Year Retrospective Cohort Study. Ophthalmol Ther. 2023. doi: 10.1007/s40123-023-00771-4.

D. Chung YR, Kim YH, Byeon HE, Jo DH, Kim JH, Lee K. Effect of a Single Intravitreal Bevacizumab Injection on Proteinuria in Patients With Diabetes. Transl Vis Sci Technol. 2020 Mar 9;9(4):4. doi: 10.1167/tvst.9.4.4.

E. Del Cura Mar P, Carballés MJC, Sastre-Ibáñez M. Risk of renal damage associated with intravitreal anti-VEGF therapy for diabetic macular edema in routine clinical practice. Indian J Ophthalmol. 2023 Aug;71(8):3091-3094. doi:10.4103/IJO.IJO_44_23.

F. Jiang L, Peng L, Zhou Y, Chen G, Zhao B, Li M, Li X. Do intravitreal anti-vascular endothelial growth factor agents lead to renal adverse events? A pharmacovigilance real-world study. Front Med (Lausanne). 2023 Feb 14;10:1100397. doi: 10.3389/fmed.2023.1100397. 

G. Neffendorf JE, Mare T, Simpson ARH, Soare C, Kirthi V, Sharpe CC, Jackson TL. Effect of intravitreal anti-vascular endothelial growth factor treatment for neovascular age-related macular degeneration on renal function. Nephrol Dial Transplant. 2023 Jun 30;38(7):1770-1772. doi: 10.1093/ndt/gfad035. 

H. Chen X, Zhang X, Gong Z, Yang Y, Zhang X, Wang Q, Wang Y, Xie R. The link between diabetic retinal and renal microvasculopathy is associated with dyslipidemia and upregulated circulating level of cytokines. Front Public Health. 2023 Jan 17;10:1040319. doi: 10.3389/fpubh.2022.1040319. 

I. Ou SH, Yin CH, Chung TL, Chen HY, Chen CL, Chen JS, Lee PT. Intravitreal Vascular Endothelial Growth Factor Inhibitor Use and Renal Function Decline in Patients with Diabetic Retinopathy. Int J Environ Res Public Health. 2022 Nov 1;19(21):14298. doi: 10.3390/ijerph192114298. 

J. Gan G, Michel M, Max A, Sujet-Perone N, Zevering Y, Vermion JC, Zaidi M, Savenkoff B, Perone JM. Membranoproliferative glomerulonephritis after intravitreal vascular growth factor inhibitor injections: A case report and review of the literature. Br J Clin Pharmacol. 2023 Jan;89(1):401-409. doi:10.1111/bcp.15558. 

Comments on the Quality of English Language

The subject the authors address, of CKD worsening during bevacizumab treatment is an important one and the authors’ finding of an epidemiological association is not a new finding but a useful one that should be published, confirming previous findings of others. 

I have a minor ethical concern that the authors did not refer to previous similar findings. Confirming previous findings of others is useful and important. Hopefully their omission was inadvertent or due to poor scholarship, not an attempt at hiding previous related work. English use is good but presentation is a bit sloppy and must be corrected. A few sentences are hard to understand and must be corrected.

Line 35 assumes readers know what bev is. Dont assume this. Better would be “Bevacizumab is a pharmaceutical monoclonal anti-VEGF antibody that…”

Line 40 a dash or an “and” missing “2012 2018”. So “between 2012 and 2018 were identified…”.

Line 47, would the authors think it helpful to list the actual numbers here ?

Line 54. Would it be helpful to add the common definition of CKD as GFR  below 60 ml/min/1.73M2 ?

Line 56 I don’t think we can speak of “serious CKD” in a medical or formal paper on renal function. I think you must define any discussion of CKD in terms of grade [1, 2, 3 or 4] and the implications each grade has. Remember you are addressing both non-clinical researchers as well as front line physicians. This is not a newpaper article.

Your sentence “Endothelial dysfunction is a significant factor contributing to the increased cardiovascular risk associated with CKD(2).” is a bit unclear or confusing. The relationship between CKD, endothelial dysfunction and cardiovascular disease [CVD] are indeed related but you must explain this more clearly. It isn't simple. Vascular pathology can lead to both CKD and to CVD. CKD can generate systemic vascular pathology and CVD. A fourth factor [chemotherapies, substance abuse, sepsis, etc] can generate all three, CKD, CVD and systemic vasculopathy. An example of this fourth potential relationship was discussed in H. Chen et al below. You must find a way to simply but clearly say this.

Sometimes there is a space between the last word and the references and sometimes there isn’t [VEGF(7-10)]. Fix this such that there is always a space.

Lines 65 to 69 should be a separate paragraph. 

Line 76, this might be an idiosyncrasy of mine, but giving these statistics to the hundredth place seems silly, obfuscating and scientifically wrong or misleading. The authors do not doubt those numbers in the hundredth place would be different if they used a different time frame for both cohorts or a different number in each cohort. Remember the word “significance has two meanings - 1) in statistics, and 2) meaningfulness to the reader. If I hand you a penny you are statistically richer but not meaningfully richer. If I hand a starving man two grains of rice by technical measurement he has more food than he did prior to that, a statistically meaningful difference, but not a meaningful difference to him. Likewise reporting “1.73±1.32 years“ is distracting and deceptive. Would not “1.7 years ± 1.3” be more meaningful and easier to read and absorb ? My similar complaint applies throughout this paper.

Lines 77 to 79 are example of above. The authors state “In the bevacizumab cohort, 58.3% were men, and the mean age was 61.53±12.49 years; 47.4% were classified as medium income and 52.7% were categorized as high urbanization level.”. My view is better would be “In the bevacizumab cohort, 58% were men, and the mean age was 62 ±12 years; 47% were classified as medium income, 53% were categorized as high urbanization level.” This would be a more effective communication and more importantly, IMO, more accurate. 

Line 79 starts a new subject, should therefore start new paragraph.

Line 80-81, CCB not previously defined. “alpha-blockers” is unacceptable [alpha-adrenergic blockers presumably], ARB not previously defined, DM not previously defined, 

Line 81, no ACE inhibitors ? in USA, Canada, and the EU lisinopril is commonly used in CKD. You didn’t have many lisinopril pts ?

Line 88, “contracting” incorrect word. “developing” is your intended meaning.

Lines 88 to 94, I can assure the authors that they obscure their message by this paragraph. First you use CI here and ± elsewhere. Are these not the same ? Also in this paragraph the use of hundredth place is utterly unwarranted and obscuring to your findings. I would also claim that use of tenth place is superfluous, distracting, and unwarranted by the nature of your cohort or your findings.  

Line 94 HTN not previously defined. Although most clinical people in English speaking countries will know what this stands for, non-clinicians may not and those in non-English speaking areas might not. This error of presentation again reflects the authors not carefully thinking of how their readers might be positioned.

Line 104 COPD not previously defined.

Line 92 ACEI not previously defined. 

Lines 96 to 108 must be rewritten. 

Your message will be lost when you try to present data this way in a maze of numbers. I suggest simply reduce this entire paragraph to….. 

“Furthermore, we conducted a stratified analysis as presented in Table 3. Compared to control patients, both males and females of bevacizumab faced a greater risk of CKD. Across all age and income groups, the risks of CKD associated with bevacizumab treatment were higher than control groups. When stratified by comorbidities and medications, patients within the bevacizumab cohort with DM, dyslipidemia, HTN, COPD, alpha-adrenergic blocker usage, ARB usage, beta blocker usage, CCB usage, diuretic usage, and other anti-HTN drugs usage exhibited an increased risk of CKD compared to the non-bevacizumab cohort in these same categories.”. 

There is no need to duplicate numbers in the tect that are listed in Table3.

I suggest leaving out the actual number of people in all Tables. Just list %.

SMD requires defining in Tables.

Table 2, better just list %. TWD needs defining. CI need not be defined. Would it help international readers to show TWD in USD or Euros ?

I don’t think Table 3 will be of much use to people. I am a physician-clinician and I think I am representative of that group. We are much more likely to wade through that maze of numbers if you eliminated actual numbers in each group and just listed %, HR and CI.

Line 125, no. CKD is not “CKD is a condition in which the kidneys are damaged causing impaired renal function.” CKD refers to a condition of GFR <60. Often, but not always, due to glomerular damage. 

Line 138 is incorrect. We see wound non-healing and HTN as the most common bevacizumab short term adverse events. 

Line 151, is incorrect. “poisonous effects” is not the same as damaging effects.

Line 155, 156 is bad grammar. Paragraph 155-158 must be rewritten. Perhaps say simply “others studying intraocular bevacizumab did not find reduced GFR”. It would be obvious why they did not. But some did and they too must be discussed

Line 167, I did not understand the authors’ meaning. [ Nevertheless, the National Health Insurance (NHI) program covers >99% of Taiwan residents and reduces the gap in medical convenience in remote areas by providing free medical care(37, 38), thus minimizing surveillance bias.] ? Please rephrase.

Line 165, new aspect. “Second, different medical…” start new paragraph.

Line 167 again shows the authors’ unclarity on what CKD incompases. Stage 1 and 2 are usually clinically silent but 3 and 4 are less so.

“closely” and “urgently” must be eliminated. GFR is always monitored during most cancer treatments and so will catch any reduced GFR. The authors could mention that understanding the details of how exactly bevacizumab damages endothelium might help us protect glomeruli during bev treatment. Also in cases of grade 3 CKD one might carefully balance r/b during cancer use. Not rare is sign/symptom improvement in cancer but no prolonging of survival with bev use is seen. One might then consider which is worse, bev or the signs/symptoms one is trying to relieve. 

References A to J must be discussed and included in Discussion Section. A diagram [a cartoon or schematic drawing] would be helpful when discussing these papers. Also the authors should review the way that these authors presented their data. Note for example that Fang et al was easy to read, omitted the hundredth place in data, and was good example of effective medical communication.

Reference 1 is intolerable in a medical scientific research article.

The entire reference list is too old, it must be redone and the text changed correspondingly to reflect current understandings of endothelium, bevacizumab and CKD. The authors’ physiology discussions are inadequate, out-of-date and must be corrected.

-------------------------------------------------------------------------

A. Li J, Li XL, Li CQ. Immunoregulation mechanism of VEGF signaling pathway inhibitors and its efficacy on the kidney. Am J Med Sci. 2023 Sep10:S0002-9629(23)01330-7. doi: 10.1016/j.amjms.2023.09.005. 

B. Zhang J, Zhu H, Tang Q, Wu J. Correlation Analysis of Vascular Endothelial Growth Factor Level with Clinicopathological Features and Prognosis in Patients with Diabetic Nephropathy: A Biopsy-based Study. Cell Mol Biol (Noisy-le-grand). 2023 Aug 31;69(8):192-197. doi: 10.14715/cmb/2023.69.8.29. 

C. Fang YC, Lai IP, Lai TT, Chen TC, Yang CH, Ho TC, Yang CM, Hsieh YT. Long-Term Change in Renal Function After Intravitreal Anti-VEGF Treatment for Diabetic Macular Edema: A 2-Year Retrospective Cohort Study. Ophthalmol Ther. 2023. doi: 10.1007/s40123-023-00771-4.

D. Chung YR, Kim YH, Byeon HE, Jo DH, Kim JH, Lee K. Effect of a Single Intravitreal Bevacizumab Injection on Proteinuria in Patients With Diabetes. Transl Vis Sci Technol. 2020 Mar 9;9(4):4. doi: 10.1167/tvst.9.4.4.

E. Del Cura Mar P, Carballés MJC, Sastre-Ibáñez M. Risk of renal damage associated with intravitreal anti-VEGF therapy for diabetic macular edema in routine clinical practice. Indian J Ophthalmol. 2023 Aug;71(8):3091-3094. doi:10.4103/IJO.IJO_44_23.

F. Jiang L, Peng L, Zhou Y, Chen G, Zhao B, Li M, Li X. Do intravitreal anti-vascular endothelial growth factor agents lead to renal adverse events? A pharmacovigilance real-world study. Front Med (Lausanne). 2023 Feb 14;10:1100397. doi: 10.3389/fmed.2023.1100397. 

G. Neffendorf JE, Mare T, Simpson ARH, Soare C, Kirthi V, Sharpe CC, Jackson TL. Effect of intravitreal anti-vascular endothelial growth factor treatment for neovascular age-related macular degeneration on renal function. Nephrol Dial Transplant. 2023 Jun 30;38(7):1770-1772. doi: 10.1093/ndt/gfad035. 

H. Chen X, Zhang X, Gong Z, Yang Y, Zhang X, Wang Q, Wang Y, Xie R. The link between diabetic retinal and renal microvasculopathy is associated with dyslipidemia and upregulated circulating level of cytokines. Front Public Health. 2023 Jan 17;10:1040319. doi: 10.3389/fpubh.2022.1040319. 

I. Ou SH, Yin CH, Chung TL, Chen HY, Chen CL, Chen JS, Lee PT. Intravitreal Vascular Endothelial Growth Factor Inhibitor Use and Renal Function Decline in Patients with Diabetic Retinopathy. Int J Environ Res Public Health. 2022 Nov 1;19(21):14298. doi: 10.3390/ijerph192114298. 

J. Gan G, Michel M, Max A, Sujet-Perone N, Zevering Y, Vermion JC, Zaidi M, Savenkoff B, Perone JM. Membranoproliferative glomerulonephritis after intravitreal vascular growth factor inhibitor injections: A case report and review of the literature. Br J Clin Pharmacol. 2023 Jan;89(1):401-409. doi:10.1111/bcp.15558. 

Author Response

he subject the authors address, of CKD worsening during bevacizumab treatment is an important one and the authors’ finding of an epidemiological association is not a new finding but a useful one that should be published, confirming previous findings of others.

I have a minor ethical concern that the authors did not refer to previous similar findings. Confirming previous findings of others is useful and important. Hopefully their omission was inadvertent or due to poor scholarship, not an attempt at hiding previous related work. English use is good but presentation is a bit sloppy and must be corrected. A few sentences are hard to understand and must be corrected.

Line 35 assumes readers know what bev is. Dont assume this. Better would be “Bevacizumab is a pharmaceutical monoclonal anti-VEGF antibody that…”

Reply

We modified the sentence with this suggestion. (line 39-40)

Line 40 a dash or an “and” missing “2012 2018”. So “between 2012 and 2018 were identified…”.

Reply

Sorry for the typo. We corrected it. (line 45)

Line 47, would the authors think it helpful to list the actual numbers here ?

Reply

We rewrote the sentence. (line 51-52)

Line 54. Would it be helpful to add the common definition of CKD as GFR below 60 ml/min/1.73M2 ?

Reply

We added the definition of CKD. (line 59-60)

Line 56 I don’t think we can speak of “serious CKD” in a medical or formal paper on renal function. I think you must define any discussion of CKD in terms of grade [1, 2, 3 or 4] and the implications each grade has. Remember you are addressing both non-clinical researchers as well as front line physicians. This is not a newpaper article.

Reply

We rewrote the sentence. (line 61)

Your sentence “Endothelial dysfunction is a significant factor contributing to the increased cardiovascular risk associated with CKD(2).” is a bit unclear or confusing. The relationship between CKD, endothelial dysfunction and cardiovascular disease [CVD] are indeed related but you must explain this more clearly. It isn't simple. Vascular pathology can lead to both CKD and to CVD. CKD can generate systemic vascular pathology and CVD. A fourth factor [chemotherapies, substance abuse, sepsis, etc] can generate all three, CKD, CVD and systemic vasculopathy. An example of this fourth potential relationship was discussed in H. Chen et al below. You must find a way to simply but clearly say this.

Reply

In the Discussion, we clearly discussed the association between VEGF and CKD in red font, and  We didn't pay attention to CVD in this study.

Sometimes there is a space between the last word and the references and sometimes there isn’t [VEGF(7-10)]. Fix this such that there is always a space.

Reply

We fixed the problem with this suggestion.

Lines 65 to 69 should be a separate paragraph.

Reply

We fixed the problem with this suggestion.

Line 76, this might be an idiosyncrasy of mine, but giving these statistics to the hundredth place seems silly, obfuscating and scientifically wrong or misleading. The authors do not doubt those numbers in the hundredth place would be different if they used a different time frame for both cohorts or a different number in each cohort. Remember the word “significance has two meanings - 1) in statistics, and 2) meaningfulness to the reader. If I hand you a penny you are statistically richer but not meaningfully richer. If I hand a starving man two grains of rice by technical measurement he has more food than he did prior to that, a statistically meaningful difference, but not a meaningful difference to him. Likewise reporting “1.73±1.32 years“ is distracting and deceptive. Would not “1.7 years ± 1.3” be more meaningful and easier to read and absorb ? My similar complaint applies throughout this paper.

Reply

We fixed the problem with this suggestion.

Lines 77 to 79 are example of above. The authors state “In the bevacizumab cohort, 58.3% were men, and the mean age was 61.53±12.49 years; 47.4% were classified as medium income and 52.7% were categorized as high urbanization level.”. My view is better would be “In the bevacizumab cohort, 58% were men, and the mean age was 62 ±12 years; 47% were classified as medium income, 53% were categorized as high urbanization level.” This would be a more effective communication and more importantly, IMO, more accurate.

Reply

We fixed the problem with this suggestion.

Line 79 starts a new subject, should therefore start new paragraph.

Reply

We fixed the problem with this suggestion.

Line 80-81, CCB not previously defined. “alpha-blockers” is unacceptable [alpha-adrenergic blockers presumably], ARB not previously defined, DM not previously defined,

Reply

We previously defined the abbreviations with this suggestion.

Line 81, no ACE inhibitors ? in USA, Canada, and the EU lisinopril is commonly used in CKD. You didn’t have many lisinopril pts ?

Reply

In Taiwan, physicians will choose between ACE inhibitors and angiotensin receptor blocker.

Line 88, “contracting” incorrect word. “developing” is your intended meaning.

Reply

We corrected it by this suggestion. (line 105)

Lines 88 to 94, I can assure the authors that they obscure their message by this paragraph. First you use CI here and ± elsewhere. Are these not the same ? Also in this paragraph the use of hundredth place is utterly unwarranted and obscuring to your findings. I would also claim that use of tenth place is superfluous, distracting, and unwarranted by the nature of your cohort or your findings. 

Reply

CI (confidence interval) and ±SD are different in statistics. The distribution of HR is not symmetric. If CI includes 1, the risk is not significantly higher/lower than the reference group. Mean±SD means the distribution is symmetric, such as distributions of age and follow-up years.

Line 94 HTN not previously defined. Although most clinical people in English speaking countries will know what this stands for, non-clinicians may not and those in non-English speaking areas might not. This error of presentation again reflects the authors not carefully thinking of how their readers might be positioned.

Reply

We previously defined the abbreviations with this suggestion.

Line 104 COPD not previously defined.

Reply

We previously defined the abbreviations with this suggestion.

Line 92 ACEI not previously defined.

Reply

We previously defined the abbreviations with this suggestion.

Lines 96 to 108 must be rewritten.

Your message will be lost when you try to present data this way in a maze of numbers. I suggest simply reduce this entire paragraph to…..

“Furthermore, we conducted a stratified analysis as presented in Table 3. Compared to control patients, both males and females of bevacizumab faced a greater risk of CKD. Across all age and income groups, the risks of CKD associated with bevacizumab treatment were higher than control groups. When stratified by comorbidities and medications, patients within the bevacizumab cohort with DM, dyslipidemia, HTN, COPD, alpha-adrenergic blocker usage, ARB usage, beta blocker usage, CCB usage, diuretic usage, and other anti-HTN drugs usage exhibited an increased risk of CKD compared to the non-bevacizumab cohort in these same categories.”. There is no need to duplicate numbers in the tect that are listed in Table3.

Reply

We rewrote the paragraph with this suggestion. (Line 113-120)

I suggest leaving out the actual number of people in all Tables. Just list %.

Reply

We modified all tables with this suggestion.

SMD requires defining in Tables.

Reply

We added the definition of SMD.

Table 2, better just list %. TWD needs defining. CI need not be defined. Would it help international readers to show TWD in USD or Euros ?

Reply

We modified table 2 with this suggestion.

I don’t think Table 3 will be of much use to people. I am a physician-clinician and I think I am representative of that group. We are much more likely to wade through that maze of numbers if you eliminated actual numbers in each group and just listed %, HR and CI.

Reply

We modified table 3 with this suggestion.

Line 125, no. CKD is not “CKD is a condition in which the kidneys are damaged causing impaired renal function.” CKD refers to a condition of GFR <60. Often, but not always, due to glomerular damage.

Reply

We rewrote the sentence with this suggestion. (line 147-149)

Line 138 is incorrect. We see wound non-healing and HTN as the most common bevacizumab short term adverse events.

Reply

We modified it with this suggestion. (line 188)

Line 151, is incorrect. “poisonous effects” is not the same as damaging effects.

Reply

We modified it with this suggestion. (line 201)

Line 155, 156 is bad grammar. Paragraph 155-158 must be rewritten. Perhaps say simply “others studying intraocular bevacizumab did not find reduced GFR”. It would be obvious why they did not. But some did and they too must be discussed

Reply

We deleted the paragraph with this suggestion.

Line 167, I did not understand the authors’ meaning. [ Nevertheless, the National Health Insurance (NHI) program covers >99% of Taiwan residents and reduces the gap in medical convenience in remote areas by providing free medical care(37, 38), thus minimizing surveillance bias.] ? Please rephrase.

Reply

Higher medical costs may decrease the intention to seek medical treatment in residents with low social status. Most of them live in low levels of urbanization resulting in the distortion prevalence of CKD. Thus, NHI program can minimize surveillance bias.

Line 165, new aspect. “Second, different medical…” start new paragraph.

Reply

We modified it with this suggestion. (line 219)

Line 167 again shows the authors’ unclarity on what CKD incompases. Stage 1 and 2 are usually clinically silent but 3 and 4 are less so.

Reply

We modified it with this suggestion. (line 224)

“closely” and “urgently” must be eliminated. GFR is always monitored during most cancer treatments and so will catch any reduced GFR. The authors could mention that understanding the details of how exactly bevacizumab damages endothelium might help us protect glomeruli during bevtreatment. Also in cases of grade 3 CKD one might carefully balance r/b during cancer use. Not rare is sign/symptom improvement in cancer but no prolonging of survival with bev use is seen. One might then consider which is worse, bev or the signs/symptoms one is trying to relieve.

Reply

We rewrote the Conclusion with this suggestion. (line 289)

References A to J must be discussed and included in Discussion Section. A diagram [a cartoon or schematic drawing] would be helpful when discussing these papers. Also the authors should review the way that these authors presented their data. Note for example that Fang et al was easy to read, omitted the hundredth place in data, and was good example of effective medical communication.

Reply

We clearly discussed and included the references A to J in Discussion Section a with this suggestion. (line 289)

Reference 1 is intolerable in a medical scientific research article.

The entire reference list is too old, it must be redone and the text changed correspondingly to reflect current understandings of endothelium, bevacizumab and CKD. The authors’ physiology discussions are inadequate, out-of-date and must be corrected.

Reply

We modified the Reference with this suggestion.

-------------------------------------------------------------------------

  1. Li J, Li XL, Li CQ. Immunoregulation mechanism of VEGF signaling pathway inhibitors and its efficacy on the kidney. Am J Med Sci. 2023 Sep10:S0002-9629(23)01330-7. doi: 10.1016/j.amjms.2023.09.005.

  1. Zhang J, Zhu H, Tang Q, Wu J. Correlation Analysis of Vascular Endothelial Growth Factor Level with Clinicopathological Features and Prognosis in Patients with Diabetic Nephropathy: A Biopsy-based Study. Cell Mol Biol (Noisy-le-grand). 2023 Aug 31;69(8):192-197. doi: 10.14715/cmb/2023.69.8.29.

  1. Fang YC, Lai IP, Lai TT, Chen TC, Yang CH, Ho TC, Yang CM, Hsieh YT. Long-Term Change in Renal Function After Intravitreal Anti-VEGF Treatment for Diabetic Macular Edema: A 2-Year Retrospective Cohort Study. Ophthalmol Ther. 2023. doi: 10.1007/s40123-023-00771-4.

  1. Chung YR, Kim YH, Byeon HE, Jo DH, Kim JH, Lee K. Effect of a Single Intravitreal Bevacizumab Injection on Proteinuria in Patients With Diabetes. Transl Vis Sci Technol. 2020 Mar 9;9(4):4. doi: 10.1167/tvst.9.4.4.

  1. Del Cura Mar P, Carballés MJC, Sastre-Ibáñez M. Risk of renal damage associated with intravitreal anti-VEGF therapy for diabetic macular edema in routine clinical practice. Indian J Ophthalmol. 2023 Aug;71(8):3091-3094. doi:10.4103/IJO.IJO_44_23.

  1. Jiang L, Peng L, Zhou Y, Chen G, Zhao B, Li M, Li X. Do intravitreal anti-vascular endothelial growth factor agents lead to renal adverse events? A pharmacovigilance real-world study. Front Med (Lausanne). 2023 Feb 14;10:1100397. doi: 10.3389/fmed.2023.1100397.

  1. Neffendorf JE, Mare T, Simpson ARH, Soare C, Kirthi V, Sharpe CC, Jackson TL. Effect of intravitreal anti-vascular endothelial growth factor treatment for neovascular age-related macular degeneration on renal function. Nephrol Dial Transplant. 2023 Jun 30;38(7):1770-1772. doi: 10.1093/ndt/gfad035.

  1. Chen X, Zhang X, Gong Z, Yang Y, Zhang X, Wang Q, Wang Y, Xie R. The link between diabetic retinal and renal microvasculopathy is associated with dyslipidemia and upregulated circulating level of cytokines. Front Public Health. 2023 Jan 17;10:1040319. doi: 10.3389/fpubh.2022.1040319.

  1. Ou SH, Yin CH, Chung TL, Chen HY, Chen CL, Chen JS, Lee PT. Intravitreal Vascular Endothelial Growth Factor Inhibitor Use and Renal Function Decline in Patients with Diabetic Retinopathy. Int J Environ Res Public Health. 2022 Nov 1;19(21):14298. doi: 10.3390/ijerph192114298.

  1. Gan G, Michel M, Max A, Sujet-Perone N, Zevering Y, Vermion JC, Zaidi M, Savenkoff B, Perone JM. Membranoproliferative glomerulonephritis after intravitreal vascular growth factor inhibitor injections: A case report and review of the literature. Br J Clin Pharmacol. 2023 Jan;89(1):401-409. doi:10.1111/bcp.15558.

Comments on the Quality of English Language

The subject the authors address, of CKD worsening during bevacizumab treatment is an important one and the authors’ finding of an epidemiological association is not a new finding but a useful one that should be published, confirming previous findings of others.

I have a minor ethical concern that the authors did not refer to previous similar findings. Confirming previous findings of others is useful and important. Hopefully their omission was inadvertent or due to poor scholarship, not an attempt at hiding previous related work. English use is good but presentation is a bit sloppy and must be corrected. A few sentences are hard to understand and must be corrected.

Line 35 assumes readers know what bev is. Dont assume this. Better would be “Bevacizumab is a pharmaceutical monoclonal anti-VEGF antibody that…”

 Reply

We modified the sentence with this suggestion. (line 39-40)

Line 40 a dash or an “and” missing “2012 2018”. So “between 2012 and 2018 were identified…”.

 Reply

Sorry for the typo. We corrected it. (line 45)

Line 47, would the authors think it helpful to list the actual numbers here ?

 Reply

We rewrote the sentence. (line 51-52)

Line 54. Would it be helpful to add the common definition of CKD as GFR  below 60 ml/min/1.73M2 ?

 Reply

We added the definition of CKD. (line 59-60)

Line 56 I don’t think we can speak of “serious CKD” in a medical or formal paper on renal function. I think you must define any discussion of CKD in terms of grade [1, 2, 3 or 4] and the implications each grade has. Remember you are addressing both non-clinical researchers as well as front line physicians. This is not a newpaper article.

 Reply

We rewrote the sentence. (line 61)

Your sentence “Endothelial dysfunction is a significant factor contributing to the increased cardiovascular risk associated with CKD(2).” is a bit unclear or confusing. The relationship between CKD, endothelial dysfunction and cardiovascular disease [CVD] are indeed related but you must explain this more clearly. It isn't simple. Vascular pathology can lead to both CKD and to CVD. CKD can generate systemic vascular pathology and CVD. A fourth factor [chemotherapies, substance abuse, sepsis, etc] can generate all three, CKD, CVD and systemic vasculopathy. An example of this fourth potential relationship was discussed in H. Chen et al below. You must find a way to simply but clearly say this.

 Reply

In the Discussion, we clearly discussed the association between VEGF and CKD in red font, and We didn't pay attention to CVD in this study.

Sometimes there is a space between the last word and the references and sometimes there isn’t [VEGF(7-10)]. Fix this such that there is always a space.

 Reply

We fixed the problem with this suggestion.

Lines 65 to 69 should be a separate paragraph.

 Reply

We fixed the problem with this suggestion.

Line 76, this might be an idiosyncrasy of mine, but giving these statistics to the hundredth place seems silly, obfuscating and scientifically wrong or misleading. The authors do not doubt those numbers in the hundredth place would be different if they used a different time frame for both cohorts or a different number in each cohort. Remember the word “significance has two meanings - 1) in statistics, and 2) meaningfulness to the reader. If I hand you a penny you are statistically richer but not meaningfully richer. If I hand a starving man two grains of rice by technical measurement he has more food than he did prior to that, a statistically meaningful difference, but not a meaningful difference to him. Likewise reporting “1.73±1.32 years“ is distracting and deceptive. Would not “1.7 years ± 1.3” be more meaningful and easier to read and absorb ? My similar complaint applies throughout this paper.

 Reply

We fixed the problem with this suggestion.

Lines 77 to 79 are example of above. The authors state “In the bevacizumab cohort, 58.3% were men, and the mean age was 61.53±12.49 years; 47.4% were classified as medium income and 52.7% were categorized as high urbanization level.”. My view is better would be “In the bevacizumab cohort, 58% were men, and the mean age was 62 ±12 years; 47% were classified as medium income, 53% were categorized as high urbanization level.” This would be a more effective communication and more importantly, IMO, more accurate.

 Reply

We fixed the problem with this suggestion.

Line 79 starts a new subject, should therefore start new paragraph.

 Reply

We fixed the problem with this suggestion.

Line 80-81, CCB not previously defined. “alpha-blockers” is unacceptable [alpha-adrenergic blockers presumably], ARB not previously defined, DM not previously defined,

 Reply

We previously defined the abbreviations with this suggestion.

Line 81, no ACE inhibitors ? in USA, Canada, and the EU lisinopril is commonly used in CKD. You didn’t have many lisinopril pts ?

 Reply

In Taiwan, physicians will choose between ACE inhibitors and angiotensin receptor blocker.

Line 88, “contracting” incorrect word. “developing” is your intended meaning.

 Reply

We corrected it by this suggestion. (line 105)

Lines 88 to 94, I can assure the authors that they obscure their message by this paragraph. First you use CI here and ± elsewhere. Are these not the same ? Also in this paragraph the use of hundredth place is utterly unwarranted and obscuring to your findings. I would also claim that use of tenth place is superfluous, distracting, and unwarranted by the nature of your cohort or your findings. 

 Reply

CI (confidence interval) and ±SD are different in statistics. The distribution of HR is not symmetric. If CI includes 1, the risk is not significantly higher/lower than the reference group. Mean±SD means the distribution is symmetric, such as distributions of age and follow-up years.

Line 94 HTN not previously defined. Although most clinical people in English speaking countries will know what this stands for, non-clinicians may not and those in non-English speaking areas might not. This error of presentation again reflects the authors not carefully thinking of how their readers might be positioned.

 Reply

We previously defined the abbreviations with this suggestion.

Line 104 COPD not previously defined.

 Reply

We previously defined the abbreviations with this suggestion.

Line 92 ACEI not previously defined.

 Reply

We previously defined the abbreviations with this suggestion.

Lines 96 to 108 must be rewritten.

Your message will be lost when you try to present data this way in a maze of numbers. I suggest simply reduce this entire paragraph to…..

“Furthermore, we conducted a stratified analysis as presented in Table 3. Compared to control patients, both males and females of bevacizumab faced a greater risk of CKD. Across all age and income groups, the risks of CKD associated with bevacizumab treatment were higher than control groups. When stratified by comorbidities and medications, patients within the bevacizumab cohort with DM, dyslipidemia, HTN, COPD, alpha-adrenergic blocker usage, ARB usage, beta blocker usage, CCB usage, diuretic usage, and other anti-HTN drugs usage exhibited an increased risk of CKD compared to the non-bevacizumab cohort in these same categories.”.

 Reply

We rewrote the paragraph with this suggestion. (Line 113-120)

There is no need to duplicate numbers in the tect that are listed in Table3.

Reply

We modified table 3 with this suggestion.

I suggest leaving out the actual number of people in all Tables. Just list %.

 Reply

We modified all tables with this suggestion.

SMD requires defining in Tables.

 Reply

We added the definition of SMD.

Table 2, better just list %. TWD needs defining. CI need not be defined. Would it help international readers to show TWD in USD or Euros ?

Reply

We modified table 2 with this suggestion.

I don’t think Table 3 will be of much use to people. I am a physician-clinician and I think I am representative of that group. We are much more likely to wade through that maze of numbers if you eliminated actual numbers in each group and just listed %, HR and CI.

 Reply

We modified table 3 with this suggestion.

Line 125, no. CKD is not “CKD is a condition in which the kidneys are damaged causing impaired renal function.” CKD refers to a condition of GFR <60. Often, but not always, due to glomerular damage.

 Reply

We rewrote the sentence with this suggestion. (line 147-149)

Line 138 is incorrect. We see wound non-healing and HTN as the most common bevacizumab short term adverse events.

 Reply

We modified it with this suggestion. (line 188)

Line 151, is incorrect. “poisonous effects” is not the same as damaging effects.

 Reply

We modified it with this suggestion. (line 201)

Line 155, 156 is bad grammar. Paragraph 155-158 must be rewritten. Perhaps say simply “others studying intraocular bevacizumab did not find reduced GFR”. It would be obvious why they did not. But some did and they too must be discussed

 Reply

We deleted the paragraph with this suggestion.

Line 167, I did not understand the authors’ meaning. [ Nevertheless, the National Health Insurance (NHI) program covers >99% of Taiwan residents and reduces the gap in medical convenience in remote areas by providing free medical care(37, 38), thus minimizing surveillance bias.] ? Please rephrase.

 Reply

Higher medical costs may decrease the intention to seek medical treatment in residents with low social status. Most of them live in low levels of urbanization resulting in the distortion prevalence of CKD. Thus, NHI program can minimize surveillance bias.

Line 165, new aspect. “Second, different medical…” start new paragraph.

 Reply

We modified it with this suggestion. (line 219)

Line 167 again shows the authors’ unclarity on what CKD incompases. Stage 1 and 2 are usually clinically silent but 3 and 4 are less so.

 Reply

We modified it with this suggestion. (line 224)

“closely” and “urgently” must be eliminated. GFR is always monitored during most cancer treatments and so will catch any reduced GFR. The authors could mention that understanding the details of how exactly bevacizumab damages endothelium might help us protect glomeruli during bev treatment. Also in cases of grade 3 CKD one might carefully balance r/b during cancer use. Not rare is sign/symptom improvement in cancer but no prolonging of survival with bev use is seen. One might then consider which is worse, bev or the signs/symptoms one is trying to relieve.

 Reply

We rewrote the Conclusion with this suggestion. (line 289)

References A to J must be discussed and included in Discussion Section. A diagram [a cartoon or schematic drawing] would be helpful when discussing these papers. Also the authors should review the way that these authors presented their data. Note for example that Fang et al was easy to read, omitted the hundredth place in data, and was good example of effective medical communication.

 Reply

We clearly discussed and included the references A to J in Discussion Section a with this suggestion. (line 289)

Reference 1 is intolerable in a medical scientific research article.

The entire reference list is too old, it must be redone and the text changed correspondingly to reflect current understandings of endothelium, bevacizumab and CKD. The authors’ physiology discussions are inadequate, out-of-date and must be corrected.

-------------------------------------------------------------------------

  1. Li J, Li XL, Li CQ. Immunoregulation mechanism of VEGF signaling pathway inhibitors and its efficacy on the kidney. Am J Med Sci. 2023 Sep10:S0002-9629(23)01330-7. doi: 10.1016/j.amjms.2023.09.005.

  1. Zhang J, Zhu H, Tang Q, Wu J. Correlation Analysis of Vascular Endothelial Growth Factor Level with Clinicopathological Features and Prognosis in Patients with Diabetic Nephropathy: A Biopsy-based Study. Cell Mol Biol (Noisy-le-grand). 2023 Aug 31;69(8):192-197. doi: 10.14715/cmb/2023.69.8.29.

  1. Fang YC, Lai IP, Lai TT, Chen TC, Yang CH, Ho TC, Yang CM, Hsieh YT. Long-Term Change in Renal Function After Intravitreal Anti-VEGF Treatment for Diabetic Macular Edema: A 2-Year Retrospective Cohort Study. Ophthalmol Ther. 2023. doi: 10.1007/s40123-023-00771-4.

  1. Chung YR, Kim YH, Byeon HE, Jo DH, Kim JH, Lee K. Effect of a Single Intravitreal Bevacizumab Injection on Proteinuria in Patients With Diabetes. Transl Vis Sci Technol. 2020 Mar 9;9(4):4. doi: 10.1167/tvst.9.4.4.

  1. Del Cura Mar P, Carballés MJC, Sastre-Ibáñez M. Risk of renal damage associated with intravitreal anti-VEGF therapy for diabetic macular edema in routine clinical practice. Indian J Ophthalmol. 2023 Aug;71(8):3091-3094. doi:10.4103/IJO.IJO_44_23.

  1. Jiang L, Peng L, Zhou Y, Chen G, Zhao B, Li M, Li X. Do intravitreal anti-vascular endothelial growth factor agents lead to renal adverse events? A pharmacovigilance real-world study. Front Med (Lausanne). 2023 Feb 14;10:1100397. doi: 10.3389/fmed.2023.1100397.

  1. Neffendorf JE, Mare T, Simpson ARH, Soare C, Kirthi V, Sharpe CC, Jackson TL. Effect of intravitreal anti-vascular endothelial growth factor treatment for neovascular age-related macular degeneration on renal function. Nephrol Dial Transplant. 2023 Jun 30;38(7):1770-1772. doi: 10.1093/ndt/gfad035.

  1. Chen X, Zhang X, Gong Z, Yang Y, Zhang X, Wang Q, Wang Y, Xie R. The link between diabetic retinal and renal microvasculopathy is associated with dyslipidemia and upregulated circulating level of cytokines. Front Public Health. 2023 Jan 17;10:1040319. doi: 10.3389/fpubh.2022.1040319.

  1. Ou SH, Yin CH, Chung TL, Chen HY, Chen CL, Chen JS, Lee PT. Intravitreal Vascular Endothelial Growth Factor Inhibitor Use and Renal Function Decline in Patients with Diabetic Retinopathy. Int J Environ Res Public Health. 2022 Nov 1;19(21):14298. doi: 10.3390/ijerph192114298.

  1. Gan G, Michel M, Max A, Sujet-Perone N, Zevering Y, Vermion JC, Zaidi M, Savenkoff B, Perone JM. Membranoproliferative glomerulonephritis after intravitreal vascular growth factor inhibitor injections: A case report and review of the literature. Br J Clin Pharmacol. 2023 Jan;89(1):401-409. doi:10.1111/bcp.15558.
